# The Impact of Lactoferrin on the Growth of Intestinal Inhabitant Bacteria

**DOI:** 10.3390/ijms20194707

**Published:** 2019-09-23

**Authors:** Alan Vega-Bautista, Mireya de la Garza, Julio César Carrero, Rafael Campos-Rodríguez, Marycarmen Godínez-Victoria, Maria Elisa Drago-Serrano

**Affiliations:** 1Departamento de Sistemas Biológicos, Universidad Autónoma Metropolitana, Unidad Xochimilco (UAM-Xochimilco), Calzada del Hueso No. 1100, CdMx 04960, Mexico; avega9713@gmail.com; 2Departamento de Biología Celular, Centro de Investigación y de Estudios Avanzados del Instituto Politécnico Nacional (CINVESTAV-IPN), Avenida IPN No. 2508, CdMx 07360, Mexico; mireya@cell.cinvestav.mx; 3Departamento de Inmunología, Instituto de Investigaciones Biomédicas, Universidad Nacional Autónoma de México (IIB-UNAM), Ciudad Universitaria, CdMx 70228, Mexico; carrero@unam.mx; 4Sección de Estudios de Posgrado e Investigación, Escuela Superior de Medicina, Instituto Politécnico Nacional (ESM-IPN), Plan de San Luis y Díaz Mirón s/n, CdMx 11340, Mexico; citlicampos@gmail.com (R.C.-R.); maric_27@yahoo.com (M.G.-V.)

**Keywords:** lactoferrin, lactoferricin, milk, lactobacilli, bifidobacteria, immunomodulation, intestinal homeostasis

## Abstract

Lactoferrin (Lf) is an iron-binding milk glycoprotein that promotes the growth of selected probiotic strains. The effect of Lf on the growth and diversification of intestinal microbiota may have an impact on several issues, including (i) strengthening the permeability of the epithelial cell monolayer, (ii) favoring the microbial antagonism that discourages the colonization and proliferation of enteric pathogens, (iii) enhancing the growth and maturation of cell-monolayer components and gut nerve fibers, and (iv) providing signals to balance the anti- and pro-inflammatory responses resulting in gut homeostasis. Given the beneficial role of probiotics, this contribution aims to review the current properties of bovine and human Lf and their derivatives in in vitro probiotic growth and Lf interplay with microbiota described in the piglet model. By using Lf as a component in pharmacological products, we may enable novel strategies that promote probiotic growth while conferring antimicrobial activity against multidrug-resistant microorganisms that cause life-threatening diseases, especially in neonates.

## 1. Introduction

Bovine and human milk contains a wide array of bioactive components, including the iron-binding glycoprotein known as lactoferrin (Lf). This protein displays antimicrobial, anti-inflammatory, and immunomodulatory activities that contribute to the maintenance of homeostasis and to the control of life-threatening diseases in the intestine of consumers, mainly in neonates [1]. As its milk counterpart, Lf present in mucosal secretions and neutrophil secondary granules displays a wide array of functions including antimicrobial activity, as well as modulatory actions on immune response, cell proliferation, and iron metabolism, among some others [2]. Lf is a mammalian monomeric molecule of approximately 80 kDa that is organized into N- and C-terminal lobes whose divergent structural features determine, in part, its exceptional multifunctional characteristics [3]. Milk is an enriched source of Lf; in addition, it becomes a source of lactoferricin (Lfcin), which is a peptide derived from the Lf N-terminus by gastric pepsin digestion that also show antimicrobial activity against pathogens [4]. Current evidence has confirmed the findings reported in early studies about the growth promotion activity of Lf on selected probiotic strains as documented in in vitro assays and in vivo experiments [5,6]. Lf and Lfcin inhibit the growth of gut-beneficial microorganisms, but intriguingly, both Lf and Lfcin enhance the growth of certain selected probiotic strains such as bifidobacteria and lactobacilli or they not at all have effects on their growth [7]. These effects may result from the physicochemical properties of probiotic surface components and secreted enzymes that favor the growth-stimulating activity of Lf over its antimicrobial effects [8,9,10,11].

At present, the intestinal impact of Lf on the maturation, function, and microbiota has been addressed in clinical trials of populations at risk of life-threatening sepsis, such as pre-term and full-term newborn neonates and infants [12]. Current assays conducted in piglets as an animal model of neonatal diseases including sepsis, have documented the in vivo effects of Lf on the growth and diversity of intestinal microbiota impact on the intestinal wellness. The benefits of Lf via microbiota may entail several mechanisms, including (i) energy supply for intestinal growth [5,13], (ii) gut-brain-microbiota axis signaling for the maturation of nerve fibers [5,14], and (iii) intestinal cell maturation and differentiation. Moreover, Lf displays intrinsic properties that modulate immune and inflammatory responses and account for the beneficial impact on the maintenance of intestinal wellness [15].

Having in mind the pivotal role of probiotics and indigenous bacteria in intestinal homeostasis, this contribution aims to review the current properties of bovine and human Lf and their derivatives on in vitro probiotic growth; the interplay of Lf with gut microbiota described in clinical trials and piglets is also included.

## 2. Lactoferrin and its Derivatives

Lactoferrin is an exclusive product of mammals and has a high degree of conservation among these species; Lf protein from human (hLf) and bovine (bLf) share 69% identity [16]. Lf is a highly cationic monomeric iron-binding glycoprotein that belongs to the transferrin family [17]. The canonical tertiary structure of Lf shows that the polypeptide chain is folded into homologous N- and C-terminal lobes linked covalently by a three-turn α-helix (Figure 1) [17]. Each N- and C-terminal lobe comprises two subdomains, N1–N2 and C1–C2, which form a deep cleft in which one ferric ion (Fe^3+^) is tightly bound in synergistic cooperation with a bicarbonate ion [17]. Reversible iron-binding properties account for the three different forms of the molecule: iron-depleted Lf (apoLf), monoferric Lf, and iron-saturated diferric Lf (holoLf) [17]. The N1-terminal subdomain of hLf and bLf contains a highly positively charged N-terminal amino acid sequence motif termed the Lfcin domain [18]. Human Lf and bLf display specific structural characteristics regarding the iron-binding sites, glycan attachment sites, and glycan types, and are shown in Table 1. These divergent features reflect, in part, the multiple signaling pathways involved in the expression and regulation of the Lf gene among mammalian species and gene polymorphisms within a species and account for its controversial multifunctional properties [3,19].

An analysis of Lf isolated from different fractions of human milk revealed a wide array of enzymatic activities with potential antimicrobial roles, including serine protease, maltase, amylase, and nuclease [20,21]. Lf is prominently an N-glycosylated protein, as shown in Table 1 [22,23,24]. bLf has a glycan content of 6.7–11.2%, with a high degree of heterogeneity, including, in order of abundance, mannose (Man), N-acetylglucosamine (GlcNAc), galactose (Gal), *N*-acetylgalactosamine (GalNAc), fucose (Fuc), and N-acetylneuraminic acid (sialic acid) [25]. In hLf, the glycan content is 11–40% with a high degree of glycan variation compared with bLf glycans. In hLf, the sugar content, in descending order, includes GlcNAc, Man, Gal, sialic acid, and Fuc [26]. Lf contains a common glycan motif, GlcNAc-GlcNAc-Man-Man-Man, a kind of hook for the incorporation of glycan units and chain elongation for building bi-, tri- and polyantennary glycans by glycan synthetase enzymes. Biantennary glycans are predominant in bLf, whereas bi-, tri- and polyantennary glycans are found in hLf [22].

Milk is the major source of Lf, and human milk has the highest concentration of this protein compared with other species of mammals. Iron saturation is approximately 11% for hLf and 13% for bLf, accounting for 0.15–3.0% of the total iron in milk [27]. In both humans [28] and cows [29], the Lf concentration in milk changes according to the stage of postpartum lactation, with higher concentrations in colostrum and steady decreases during the middle and late stages of milk maturation, as depicted in Table 2. The bovine Lf concentration is dependent on the cow breed and increases during inflammation associated with intramammary infections [30,31]. Fluctuations in both the extent and profile of glycosylation in Lf present in milk are found in different stages of lactation [27,32]. For example, in bLf, the N128 site of the N-lobe is 30% glycosylated in colostrum and 15% glycosylated in mature milk [23,33].

**Table 1 ijms-20-04707-t001:** Features of human and bovine lactoferrin.

	Human Lactoferrin	Bovine Lactoferrin
Iron binding sites at interface between the two N domains	Y93, Y193, H252, D61 [17]	Y92, Y192, H253, D60 [34]
Iron binding sites at interface between the two C domains	Y447, Y540, H609, D407 [17]	Y433, Y526, H595, D395 [34]
Glycan attachment sites with N-X-S/T sequence motifs	N137 (N2), N478(C2), N623 (C1) (unglycosylated) [17]	bLf -a: N281(N1), N233(N2), N368(C1), N476(C2), N545(C2) bLf -b: all except N281 [33]
Glycan type	*N*-Acetylglucosamine, fucose, galactose, mannose neuraminic acid (sialic acid); heterogeneous bi-antennary, and poly-antennary glycans [22]	*N*-acetyl-glucosamine, fucose, galactose, neuraminic acid, N-acetyl-lactosamine, mannose; heterogeneous bi-antennary glycans [22]

Several studies have characterized the properties of Lf after cleavage by proteolytic enzymes, which yield peptide derivatives with antibacterial and even probiotic growth activity [7,34]. Enzymatic degradation of Lf by the gastric enzyme trypsin is proportional to the extent of iron saturation [35] and produces an N1-terminal-derived peptide without iron-binding activity termed Lfcin (Figure 1) [36]. Bovine Lfcin (bLfcin) is a peptide derived from the bLf N-terminus that consists of a 25-residue sequence corresponding to N17–41 of bLf [37]. Human Lfcin (hLfcin) was first described as a fragment corresponding to N1–47 of full-length Lf, with two subunits, namely, fragments 1–11 and 12–47, connected by a disulfide bridge [38]. The above findings indicate that species-specific patterns of glycosylation that change according to the stage of the milk maturation may account for the divergent impact of Lf and its peptide derivatives of human or bovine origin on probiotic growth, as is described in the following sections. Accordingly, divergent patterns of glycosylation and fluctuations in Lf concentration according to the stage of maternal milk maturation seem to be critical in providing favorable conditions for the establishment of microbiota to favor maturation and growth and ultimately homeostasis in the intestine during the breast-feeding period in neonates.

## 3. Probiotics and the Modulation of the Gut Immune Response

Probiotics are not passive commensal microorganisms but, on the contrary, a strong interaction is maintained between them and the host cells. Since Lf modulates the composition, abundance, and balance of probiotics in the human intestine and in animal models (see Section 4 and Section 5), and that probiotics, in turn, modulate many aspects of the immunity of the intestinal mucosa, in this section we deal in a general way with the knowledge about the latter, on the understanding that the consumption of Lf could lead to the modulation of the intestine’s immunity through probiotics.

### 3.1. General Properties of Probiotics

Selected lactobacilli and bifidobacteria strains are commercially used in fermented food dairy products such as probiotics, defined as any live organism that is beneficial for human or animal health [39]. Probiotics survive transiently in the intestinal tract; therefore, their ingestion in dietary supplements strengthens the beneficial role of the gut microbiota [39]. Lactobacilli and bifidobacteria are Gram-positive bacteria that form part of the group of lactic acid bacteria [40]; indigenous bifidobacteria preferably colonize the caecum and colon, while lactobacilli colonize throughout the intestinal tract [41]; both generate lactic acid as an end product of fermentation by divergent biochemical pathways [40]. Bifidobacteria accomplish hexose [glucose (Glc) and fructose (Fru)] fermentation via the “bifid shunt” involving fructose-6-phosphoketolase [40]; hexose fermentation by this via generates more ATP than that generated by glycolysis in the Embden–Meyerhof–Parnas (EMP) pathway [40]. On the other hand, lactobacilli are classified as (i) obligatory homofermentative lactobacilli, which generate only lactic acid by the EMP pathway; (ii) facultative heterofermentative lactobacilli, which generate lactic acid as the major product and carbon dioxide (CO_2_) and ethanol; and (iii) obligatory heterofermentative lactobacilli, which degrade pentoses and hexoses via the 6-phosphogluconate pathway to produce lactic acid, ethanol, acetic acid, and CO_2_ [40].

Among bifidobacteria, *Bifidobacterium longum* subsp. *infantis* (*B*. *infantis*) is able to use a wide array of glycans and oligosaccharides from human milk [42], whereas all strains ferment monomeric oligosaccharides, i.e., Glc and Gal, only *B*. *infantis* and *B. breve* ferment GlcNAc, Fuc, and sialic acid [42]. Interestingly, *B. bifidum* “shares” degraded milk oligosaccharides to support the growth of other bifidobacteria species/subspecies [8]. In addition, coculture assays have shown that the amylolytic ability of *B*. *breve* produces compounds that enhance the growth of non-amylolytic *L*. *paracasei* [43]. In return, lactobacilli ferment monosaccharides to produce 1,4-dyhydroxi-2-naphtoic acid with potent bifidogenic effects on *B*. *infantis*, *B*. *bifidum,* and *B*. *breve* [44]. The ability to ferment milk oligosaccharides is diverse among lactobacilli [45], which degrade GlcNAc and Fuc, although their ability to ferment complex glycans and oligosaccharides is limited [45].

Some bifidobacterial strains have the capacity to degrade complex milk oligosaccharides, resulting in the release of short-chain fatty acids (SCFA) used for the conversion of butyrate and some other metabolites by secondary scavengers [46]. Lactobacilli and bifidobacteria are equipped with divergent machinery for sugar breakdown; while lactobacilli use simple sugars as the preferred carbon source, bifidobacteria preferentially degrade complex glycans. These may reflect the complex interplay between bifidobacteria as primary degraders of complex oligosaccharides and lactobacilli as secondary degraders of simple monosaccharides to support their mutual growth and ultimately their probiotic effect, providing intestinal benefits.

### 3.2. The Mechanisms of Intestinal Immunomodulation

Several mechanisms underlie the health-promoting effects of probiotics, including metabolic products such as SCFA as key energy sources for host cells; probiotics have a role in toxin neutralization, microbial antagonism, and upregulation of antimicrobial peptides by cell-monolayer components as well as a buffering effect on the surrounding milieu [39,47,48]. Probiotics display direct effects by producing bacteriocins, but they also promote intestinal health by leading the regulation of all players of gut barrier function, including subepithelial immune cells (T lymphocytes and dendritic cells (DCs), luminal mucus and epithelial monolayers [49,50].

#### 3.2.1. The Modulatory Effects of Probiotics on Immune Cells

Intestinal probiotic strains finely modulate cytokine production from mucosal immune cells and enterocytes. Probiotics can induce the release of proinflammatory cytokines and chemokines that enhance resistance to enteric pathogens or in contrast can also induce anti-inflammatory/regulatory cytokines that downregulate the exacerbated inflammation produced by pathogens and commensal bacteria under conditions of dysbiosis or when the integrity of the intestinal epithelium has been compromised [51]. This dual role emphasizes the importance of probiotics in the subtle modulation of the immune response to avoid intestinal tissue damage. The underlying mechanism is related to the ability of probiotics to activate different immune cells (DCs, monocytes/macrophages, and lymphocytes), thereby inducing the release of certain patterns of cytokines that can lead to the promotion of polarized Th1, Th2, or Th17 responses [52]

As an example, in the context of eubiosis, *Bacteroides* (*B*.) *fragilis* coated with polysaccharide A activates DCs, resulting in the release of transformin growth factor (TGF)-β that in turn drives the interleukin (IL)-10 generation by Forkhead box P3 (FoxP3)+ T regulatory (Treg) cells that suppressed the proinflammatory Th17 and Th1 cell responses [53]. Under conditions of dysbiosis, the modulating role of *B*. *fragilis* via FoxP3+ Treg cells is overtaken by the pro-inflammatory effect of segmented filamentous bacteria. The latter diverts the activation of DCs for the release of IL-23 and IL-6 that in turn, drive the generation of IL-17 by Th17 cells. In another study, the orally administered probiotic strains induced the development of Th17 cells to ameliorate the clinical symptoms of intestinal inflammation [54]. Interestingly, some probiotics can induce the maturation/differentiation of DCs to a pro-inflammatory phenotype (CD11c+CD11b-CD8a+) that confers protection against rotavirus infection in mice [55]; by contrast, other probiotics can induce regulatory DCs that produce high levels of anti-inflammatory/regulatory cytokines (IL-10 and TGF-β1) that decrease inflammation as described in several experimental inflammatory diseases [56]. Consequently, a careful selection of the probiotic for specific cases must be important for therapeutic consideration.

The activation of DCs by lactic acid bacteria can also lead to the activation of natural killer cells (NKCs) [57]. Oral administration of *Bacillus* (*B*.) *polyfermenticus* in humans increased the number of NKCs by 35% and the number of CD4+ and CD8+ cells [58,59] and in other human trials, probiotic intake increased the number and activity of NKCs with tumoricidal activity [58]. In addition to enhance the cellular immune response, probiotic bacteria such as *L. rhamnosus*, *L. plantarum*, *L. acidophilus*, and *L. casei* can promote the proliferation of immunoglobulin A (IgA)-plasmatic cells from the *lamina propria* in a dose-dependent manner, thereby favoring the production of secretory IgA antibodies [60]. Together, these results lead us to say that intestinal cells are continuously stimulated by the probiotic activity trying to equilibrate the exacerbated inflammatory reaction caused by pathogens or by chronic dysfunction.

#### 3.2.2. The Modulatory Effects of Probiotics on Luminal and Epithelial Gut Barrier Components

The gut epithelium constitutes the main physical barrier between the host immune system and the luminal microbiota. These cells are bridged together by intercellular junctions, such as tight and *adherens* junctions and desmosomes [61].

Notably, probiotics have been shown to regulate the expression and cellular localization of tight junction proteins, thereby making an important contribution to the maintenance of epithelial barrier integrity. The mechanism of action has been suggested to include the activation of Toll like receptor (TLR)-2-mediated signaling, which results in overexpression of the *zonula occludens* proteins. However, other cellular pathways activated by probiotics are also involved [50]. *Bacteroides thetaiotaomicron* has been documented to upmodulate the capillary network via Paneth cell signaling [62]. These results support the role of microbiota members in the generation of antimicrobial peptides and angiogenesis via Paneth cells [62] but also by promoting angiogenesis [47]. The findings indicate that the beneficial impact of probiotics may entail their modulatory properties upon tight junction proteins of the epithelial cell monolayer via innate signaling pathways and angiogenesis via Paneth cells with a critical role in gut homeostasis.

Probiotics also influence mucus secretion by goblet cells. The probiotic strains *Lactobacillus (L.) plantarum* 299v, *L. casei* GG, and *Escherichia* (*E). coli* Nissle 1917 and the probiotic mixture VSL#3 increased mucin (MUC)-2 and MUC3 gene/protein expression in HT-29 and Caco-2 cells, reducing the adherence or translocation of enteric pathogenic bacteria. Increased expression of MUC1, MUC5A, and MUC5AC was also observed in some cases [50]. Additional in vivo evidence of increased mucin in the intestinal lumen of rats was obtained after the ingestion of a probiotic mixture (VSL#3), suggesting that the upregulation of mucus secretion occurs after oral probiotic administration [63]. Accordingly, probiotics strengthen gut barrier function by modulating the mucus layer to avoid the direct contact of microbiota with the surface of the intestinal monolayer, preventing a potential proinflammatory response.

Finally, probiotics can exclude pathogenic bacteria by exerting antimicrobial effects through the secretion of peptides and other molecules. Studies using several *Lactobacillus* spp. (*L. plantarum*, *L. salivarius*, *L. brevis*, *L. johnsonii* and *L. fermentum*) have observed the production of antimicrobial peptides, such as brevican and bacteriocin-like molecules and, in some cases, hydrogen peroxide or lactic acid, which affect the in vitro and in vivo growth of several pathogenic bacteria or fungi, including *Listeria monocytogenes*, *S.* Typhimurium, *Streptococcus mutans* and *Candida albicans* [50]. Another form of action was recently described for the probiotic strain *L. helveticus* MIMLh5, which prevented the infection of pharyngeal cells with *S. pyogenes* by inhibiting the adhesion of the bacteria [64].

According to all above, the interplay between probiotics and the host provides mutual benefits, i.e., probiotics contribute to the host wellness by their wide array of modulatory actions on the components of innate and adaptive immunity, while the host provides the optimal milieu for probiotic growth. It is at this point that treatment with Lf could efficiently contribute to maintaining intestinal homeostasis through its various effects on probiotics, as described below.

## 4. Lactoferrin: Effects on Probiotic Growth

### 4.1. Bulk Milk

Several studies have analyzed the effects of bulk milk samples of bovine and human origin on the growth of probiotic bacteria. In cultures of *B. infantis*, *B. bifidum* subsp. Pennsylvanicus (*B.* Pennsylvanicus) and *B. longum,* growth-promoting activity was found higher in human milk than in cow or goat milk [65,66]. However, *B*. *bifidum, B. infantis,* and *B. breve* growth were similarly promoted by human and cow milk [66]. Human milk samples collected from one to six months postpartum had very variable growth-promoting effects on *B. infantis*, *B. breve,* and *B.* Pennsylvanicus, whereas steady bifidogenic activity was observed in cow bulk milk from tanks of different farms [67]. These results may reflect (i) the divergent fermentability of each bifidobacterial strain, (ii) the fluctuations of bioactive components in milk (such as Lf) in different postpartum stages and (iii) changes in human and cow milk composition according to the genetic background. By contrast, the inhibitory effects of human breast milk on *B. breve,* as well as on neonatal pathogens, including *E. coli* and *Staphylococcus S. epidermidis*, have been documented. In regard to the pasteurized counterpart, unpasteurized milk showed a stronger inhibitory effect on the bacterial strains, including *B. breve,* and this effect was enhanced by the addition of hLf compared with that of bLf [68]. These findings indicate the inhibitory impact of pasteurization temperature on the antimicrobial properties of Lf and evidence the inhibitory effect of hLf on some probiotic strains.

An analysis of milk fractions separated by ultrafiltration demonstrated that, in human milk, the bifidogenic activity of the nonprotein fraction (containing glycans, among other components) was greater than that of the protein fraction for *B*. Pennsylvanicus, *B. bifidum,* and *B*. *longum* [67]. By contrast, the nonprotein and protein fractions of cow milk had similar growth-promoting effects on *B*. *infantis*, *B*. *bifidum*, and *B*. *breve* [67]. These findings may reflect the divergent composition of glycans and proteins in human and cow milk. Unlike cow milk, human milk contains lacto-N-biose (LNB), which is used as a substrate by bifidobacteria capable of using complex human milk oligosaccharides, such as *B*. *infantis* [69]. An analysis of fractioned samples indicated that bulk milk displayed a higher degree of probiotic activity than milk protein fractions tested individually [65]. The bifidogenic properties of milk were related to the content of bioactive milk proteins, although the activity of casein was unclear [66,67,70]. Both Lf and α-lactalbumin isolated from cow milk displayed stronger growth-promoting effects on *B. infantis* and *B. breve* than on *B. bifidum* strains [67]. For *B. infantis,* the bifidogenic activity of a mixture containing Lf, lactoperoxidase, and lysozyme was higher than the activities of the single components tested separately [65]. Bifidogenic activity for *B. infantis* and *B. breve* was observed for Lf from mature milk but not Lf from colostrum [71]. Lf present in milk is a glycosylated protein that contains β-*N*-glycans used for bifidobacterial growth [10,22,70]. These results suggest a role for the bifidogenic activity of Lf as a “provider” of N-glycans that increase the expression of genes required for their utilization [10]. In turn, β-glycans induce the gene expression and translation of proteins (ATP-binding cassette (ABC) and phosphodiesterase proteins) involved in the transport of sugars and their concomitant incorporation into the bifid shunt fermentative pathway [72]. These findings suggest that the individual activity of each protein is synergized to enhance the growth of probiotics and is dependent on milk stage maturation, which, in turn, impacts the extent of glycosylation of Lf.

### 4.2. Apo and HoloLactoferrin

Several studies have shown that, depending on the iron saturation, Lf displays divergent effects on probiotic growth (increase, decrease, or no effect) and inhibitory activity against pathogens. Assays in probiotic cultures indicated that apobLf inhibited or did not affect the growth of lactobacilli and bifidobacterial strains [73]. By contrast, apobLf displayed an inhibitory effect on the growth of foodborne pathogens, including *S.* Typhimurium, and *Enterococcus fecalis* [73]. The selective inhibitory action of apobLf was also found on the growth of pathogens, with no effects on the growth of lactobacilli [74]. Neutralized cell-free supernatants (to deplete the effect of lactic acid) from *L*. *reuteri* or *L*. *fermentum* cultures enhanced the inhibitory activity of apobLf against pathogenic strains and even against methicillin-resistant *Staphylococcus* (S.) *aureus* (MRSA) without affecting lactobacilli growth [74,75]. A presumable mechanism involves the ability of apoLf to facilitate the entrance of the secreted antimicrobial compounds released by the probiotic strains in the culture supernatant within the MRSA bacteria [75].

The antimicrobial action of iron-depleted bLf can also be seen against protozoan like *Entamoeba histolytica* trophozoites [76]. These findings indicate that iron-free Lf either has an antimicrobial effect against pathogenic and probiotic bacteria or displays a selective growth-promoting activity on some probiotic strains.

Additional evidence showed that apoLf (both bovine and human) inhibited the growth of *B*. *infantis*, *B*. *bifidum* and *L*. *acidophilus*, whereas 66% iron-saturated bLf (holo66bLf) inhibited the growth of bifidobacteria but not lactobacilli [77]. Conversely, both holo98bLf and holo98hLf inhibited the growth of lactobacilli but not bifidobacteria. In single-culture assays, the growth of foodborne pathogens was decreased by apobLf, apohLf, and holo66bLf but was unaltered by holo98bLf. In coculture assays, apobLf and holo66bLf selectively retarded *E. coli* O157:H7 growth without affecting *B*. *infantis* growth [77]. An analysis of metal-bound Lf forms demonstrated that apobLf, Cu-bLf, and Fe-bLf enhanced the growth of bifidobacterial strains [78]. Moreover, Cu-bLf and Zn-bLf had stronger inhibitory action than apobLf on the growth of pathogenic strains of *E. coli* and *S. aureus* [78]. The bactericidal action of apobLf and Cu-bLf was strong against *E*. *coli* and was weak against *B. breve*, whereas the inhibitory or bactericidal action of holobLf was weak or absent [78].

These results support that, unlike the iron-saturated form, iron-depleted bLf limits the growth of multidrug-resistant pathogens without affecting the proliferation of some probiotics and contributes to the natural innate mechanism for maintaining the microbiota balance and gut homeostasis.

Other experimental settings documented that holobLf but not apobLf enhanced the growth of *L. acidophilus*. Moreover, both holobLf and apobLf stimulated the growth of *B*. *breve*, *B*. *infantis*, and *B*. *bifidum*, although they did not have an effect on *B*. *longum* [79]. Under culture conditions of iron deprivation, the growth of *B. breve* was enhanced by holohLf and was inhibited by apohLf [80]. The effects of iron deprivation or supplementation on growth promotion appear to be specific to each probiotic strain and Lf source. A possible mechanism is that Lf in the iron-loaded form provides iron as an essential factor for the proper function of the enzymes involved in iron reduction, DNA replication, and repair and for ABC glycan transporters (cell membrane permease) in some bifidobacterial strains [81,82]. The effects of Lf on growth modulation by supplying or depleting iron for probiotic growth are influenced by additional environmental conditions as described below.

### 4.3. Probiotic Culture Conditions

The inhibitory and stimulatory effects of Lf on the growth of probiotics have been tested under various conditions of aerobiosis, anaerobiosis, time, temperature, and iron depletion in cultures [6,71,80,83]. Experimental results have indicated that the inhibitory concentration-dependent effect of bLf (iron less than 15 mg/100 g of protein) was similar under aerobic or anaerobic conditions [83]. Under aerobic conditions, bLf had a greater inhibitory effect on the growth of pathogenic strains than on probiotic strains but displayed strong growth-promoting effects on *L. rhamnosus* ATCC 7469 and *L*. *acidophilus* BCRC 14065 in a dose-independent manner [83]. Assays examining the effect of temperature indicated that bLf showed inconsistent probiotic activity at 37 °C; at 22 °C, the growth of probiotic strains (*B. breve* and lactobacilli strains) was selectively retarded, but the addition of bLf resumed probiotic proliferation in a dose-dependent manner [6]. These findings provide insights into the development of conditions for the selective growth of probiotic strains by controlling the temperature and bLf dosage. Other experimental in vitro settings in cultures of *Lactococcus* (L.) *lactis* subsp. cremoris JCN20076 evidenced that growth promotion activity was found higher with heat-treated bLf at 65 °C for 30 min followed by heat-treated bLf at 80 °C for 5 min while unheated bLf showed less activity on *L. lactis* subsp. cremoris growth [84]. According to these findings, the impact of heating on the potentiation of growth promotion activity of bLf is a critical advantage given that the manufacturing of infant formulas entails a thermal process of pasteurization.

In iron-free cultures, the growth-promoting effects of holohLf on *B. brevis* resulted from the direct extraction of iron-bound Lf and eventual uptake by probiotic bacteria [80]. By contrast, in other assays, the probiotic activity of bovine and human Lf was not influenced by the degree of iron saturation [71]. These discoveries suggest that environmental factors may affect the extent of iron saturation of Lf and, in turn, its ability to enhance or decrease the growth of probiotics that express Lf-binding proteins, as described in the following section.

### 4.4. Probiotic Lactoferrin-Binding Proteins

Several experiments have accounted for the impact of the interaction of Lf with probiotic surface components on Lf activity. A Western blot analysis of the protein fractions of sonicates from *B*. *bifidum* indicated the presence of bLf-binding proteins in both the membrane (69 kDa) and cytosolic fractions (20, 35, 50, and 66 kDa) [85]. These experiments used biotinylated apobLf and streptavidin-avidin-labeled horseradish peroxidase. Other assays have demonstrated the growth-promoting activity of both apo- and iron-loaded bLf forms in probiotic strains expressing membrane Lf-binding proteins, including *L*. *acidophilus* (21, 41, and 67 kDa), *B. breve,* and *B. bifidum* (both 69 kDa), and *B. infantis* (67 kDa) but not *B*. *longum* [79]. The probiotic activity of bLf isolated from mature milk was coincident with a strong degree of binding of *B. bifidum* and *B. breve*. The underlying mechanisms are apparently independent of bLf-iron saturation extent, although probiotic growth might result from the binding of Lf with probiotic proteins to be transported to the inner milieu for the concomitant cleavage and transport of Lf-linked glycans used as energy sources, as depicted in Figure 2.

In bifidobacteria, surface proteins involved in the hydrolysis and uptake of glycans (intact human milk oligosaccharides, lacto-N-biose (LNB), and free monosaccharides) are ultimately funneled to the fermentative bifid shunt pathway [69,72]. Bifidobacteria express membrane proteins related to metabolism and proteins involved in energy production and conversion [86]. Other experimental assays demonstrated that bLf from colostrum could bind *B*. *bifidum* and *B*. *breve* despite lacking growth-promoting activity. These findings should indicate that the bifidogenic properties of bLf are independent of the receptor binding capacity [71]. Visualization by confocal laser-scanning microscopy using biotinylated Lf and fluorescein-conjugated avidin showed Lf binding to the pole of *Bifidobacterium* cells [87,88]. A Lf-binding protein with a molecular mass of 67 kDa was found in the membrane and cytosolic fractions of bifidobacterial cells, including *B. longum* [87,88]. An analysis of each Lf lobe indicated that the N-lobe was the presumable site of interaction of full-length Lf with bifidobacterial proteins; by contrast, the C-lobe lacked the binding activity to bifidobacterial proteins despite promoting bifidobacterial growth [89]. A potential mechanism underlying these findings may involve ionic interactions of cationic Lf with anionic surface proteins that enable Lf uptake by bifidobacteria and subsequent potential growth-promoting activity. Human milk Lf forms a molecular complex with ATP via the C-lobe that leads to its dissociation into monomeric forms that interact with macromolecules such as DNA [90]. Human milk Lf includes enzymes such as malto-oligosaccharidase, ATPase, DNase, RNase, and phosphatase [21]. Lf acts as a DNA transcriptional factor and as an ATP-binding protein that results in the dissociation of monomeric forms that interact with DNA [91]. Future studies may provide substantive evidence that supports the probiotic effect of Lf by acting as a DNA transcriptional factor.

By contrast, hydrophobicity and autoaggregation are physicochemical parameters that reflect the ability of bacteria to interact with components of the surrounding environment [92]. In this regard, assays of bifidobacterial cells in liquid media demonstrated that autoaggregation was reduced by bLf, bovine transferrin, and ovotransferrin. Surface hydrophobicity was altered by bovine transferrin but not by bLf or Tf [9]. These results may reflect the wide array of the extent of aggregation and surface hydrophobicity, which affects the plasticity of the adaptation of bifidobacteria in the intestinal milieu [93,94].

### 4.5. Lactoferrin Hydrolysates and Lf-Derived Peptides

Although their actual mechanism of action is not fully known, the activity of Lf hydrolysates and Lf peptides on the growth of lactic acid bacteria is iron-independent, since these derivatives are unable to bind iron. Culture assays in bifidobacteria and lactobacilli strains showed that bacterial growth was either inhibited or unaltered by the hydrolysate of apobLf obtained by treatment with pepsin [73]. Moreover, compared with the full-length protein, the apobLf hydrolysate displayed strong inhibitory action on the growth of foodborne pathogens and MRSA [73,75]. Combining culture supernatants from apobLf-resistant probiotic strains with apobLf hydrolysate resulted in synergistic inhibitory action against foodborne pathogens and MRSA [73,75]. These results may reflect the role of the synergistic antimicrobial effect of bLf hydrolysate and probiotic strains resistant to bLf hydrolysate in the intestinal milieu. Stronger antimicrobial activity of apoLf hydrolysate than the full-length Lf protein may rely on its ability to enhance the entrance of the secreted antimicrobial compounds released by the probiotic strains in the culture supernatant within the MRSA bacteria [73,75]. In contrast to this inhibitory activity, bLf hydrolysates exhibited growth-promoting properties on bifidobacterial strains [95]. The bifidogenic activity on *B. breve* was greater with apobLf (pepsin) hydrolysate than with full-length Lf, while on *B. infantis*, bifidogenic activity was only found for apobLf hydrolysate. Assays on *B. breve* and *B. longum subsp. infantis* cultures indicated that a synthetic peptide from bLfcin showed stronger bifidogenic activity than natural bLf hydrolysate, whereas no bifidogenic activity was detected on an amino acid mixture containing the same amino acid sequences as the synthetic active peptide [95]. According to these effects, bLf hydrolysates and synthetic bLf peptides present in maternal milk or even in infant formula may provide intestinal benefits in infants, in addition to the benefits given by the whole Lf protein.

In human milk subjected to pepsin proteolysis, peptide derivatives from the polymeric immunoglobulin receptor (pIgR) and hLf enhanced the growth of *B*. *bifidum*. Treatment with proteolytic enzymes caused no loss of the bifidogenic activity of these peptides. A synthetic peptide derivative C(1)AV GGG CIAL(10) with a sulfide bridge in C(1) and C(7) displayed the same activity as native peptides [95]. These interesting results may explain the ability of degradation of pIgR and Lf to render active peptides refractory to enzymatic hydrolysis, promote the growth of bifidogenic cells, and inhibit the proliferation of pathogenic bacteria in the large intestine of infants [96]. In addition, the products of proteolysis of Lf by lactic acid bacteria peptidases have been characterized. Compared to other milk-derived peptides that underwent full hydrolysis, the 25-residue peptide Lfcin was more resistant to proteolytic degradation by these peptidases during incubation with *Streptococcus thermophilus* and *L. delbrueckii* subsp. *bulgaricus* strains used in the yogurt-making industry [97]. The biological activity was observed in peptide sequences located in the C-lobe of Lf. Synthetic N-L-N-R (C-lobe residues 563–566) enhanced the growth of *L. acidophilus* used to produce fermented milk while displaying strong antibacterial activity against *Pseudomonas spp.* and *E*. *coli* strains that reduce milk quality [98]. Based on these results, some hypothetical mechanisms that may account for the effects of Lf on in vitro probiotic growth are depicted in Figure 2.

## 5. Lactoferrin: Modulatory Effects on the Gut Microbiota

### 5.1. Human neonates

Importantly, it has been demonstrated that hLf benefits the nascent gut health and immune development and functioning in preterm and neonate infants. These effects are due to Lf favors the decrease of its permeability and increase of its maturation [99]. The impact of Lf on the colonization of intestinal microbiota has also been addressed in neonate populations, since ongoing intestinal maturation puts them at risk for life-threatening diseases, including necrotizing enterocolitis [100,101]. Buccigrossi et al. (2007) found that both hLf and bLf exert a mucosal trophic effect on enterocytes (Caco2 cells) that is related to its concentration; at high Lf concentrations, it was promoted a more rapid proliferation of these cells, whereas at low Lf concentration it was induced their differentiation [102].

The role of Lf on the colonization of microbiota has been addressed during the early stages of intestinal maturation in neonates fed maternal milk and an infant formula diet [100]. In early trials, a relationship between the bLf contained in an infant formula diet and the fecal microbiota in neonates was not seen [103,104]. In current studies, the abundance of fecal bifidobacteria and lactobacilli was significantly associated with the levels of Lf in feces from breastfed newborns [105]. These discoveries suggest that hLf levels in neonates are beneficial for contributing to the establishment of the gut microbiota.

Sherman et al. (2004) studied the early colonization of the immature small intestine by *Lactobacillus rhamnosus* (LGG); in addition, they explored the use of recombinant hLf (rhLf) to promote growth of LGG and to enhance gut defenses against a pathogenic *Escherichia (E.)*. *coli* strain that infects the small bowel [106]. In the experiment with newborn rat pups treated with intragastric rhLf plus LGG, they quantified gut colonization by this species. Control pups initially had lactic acid bacteria that colonized the bowel, but these bacteria were not LGG. Pups treated with LGG or rhLF plus LGG had significantly higher numbers of LGG in the ileum versus jejunum. The authors concluded that prophylactic therapy with rhLf and LGG acts to enhance defenses against pathogenic *E. coli* strain in the nascent small intestine, and suggest that rhLf and LGG are therapeutic agents that may reduce necrotizing enterocolitis (NEC) and gut-related sepsis in preterm human infants.

Manzoni et al. (2009) have been pioneers in studying the role of bLf and probiotics in the prevention of late onset of fungal and bacterial diseases in very-low birth weight (VLBW) preterm infants [107]. They designed a prospective, multicenter, double-blind, randomized trial in 11 Italian tertiary neonatal intensive care units. Patients were 472 VLBW infants; one group received orally 100 mg/day of bLf, another with bLf plus 6 × 10^9^ cfu of LGG, and the placebo group, for 30 days. The researchers concluded that bLf supplementation, alone or in combination with LGG, reduced the incidence of a first episode of late-onset sepsis in VLBW neonates. No adverse effects due to bLf were observed.

Regarding the microbiota composition and its relation with Lf intake, Mastromarino et al. (2014) measured the content of Lf and the microbiota of breast milk and of feces of infants at birth and one month after delivery [105]. Interestingly, in preterm infants, higher concentrations of fecal Lf at birth and 30 days after delivery were observed than in full-term infants; also, the amount of fecal bifidobacteria and lactobacilli were significantly associated with the concentration of fecal Lf. These results suggested that Lf promotes a bifidogenic microflora in the gut in neonate and preterm infants. High levels of fecal Lf in in the first days of life contribute to a strong early host-microbe interaction that could be important for the composition of the neonatal gut microbiota and the development of these microorganisms, in addition to the antimicrobial activity of this milk glycoprotein. This interaction is critical for having a healthy immune system and a correct metabolic program in newborns.

A clinical trial in VLBW (<1500 g) babies tested the effect of a rrhLf, administered at 150 mg/kg/day every 12 h by a nasogastric route from day one to 28 of life. *Proteobacteria* and *Firmicutes* were the major *Phyla* in feces from babies treated or untreated (placebo) with rhLf. It should be noted that the fecal abundance of the pathogenic species *Enterobacter*, *Klebsiella,* and *Staphylococcus* was decreased, while *Citrobacter* abundance was increased, but it was not associated with infections [108]. In breastfed babies, the abundance of fecal bifidobacteria was predominant, and that of the facultative anaerobes was poor; by contrast, in babies fed formula containing bLf (10 or 100 mg/ml), obligate anaerobes (*Clostridium* and *Bacteroides*) were predominantly seen in regard to bifidobacteria [109]. A delicate balance of microbiota members is prone to the disturbance in preterm infants [110]; thus, findings indicate that the effect of bLf dosage on promoting anaerobic growth must be kept in mind to prevent potential risks in dysbiosis in newborns.

In a retrospective cohort study in preterm babies fed bLf in combination with LGG, the incidence of severe necrotizing colitis was significantly reduced, and the resolution of this disease was improved. Notably, bLf had no collateral effects, but a severe case of sepsis by LGG was found [111]. These results indicate that due to the extreme fragility of very low birth weight neonates, stringent precautions should be taken to avoid the risk of severe cases of neonatal sepsis.

Currently, several recommendations of these therapies using Lf and probiotics against NEC and other gut diseases have been addressed. For example, the therapy with Lf must be as early in the life as possible, more than 100 mg/day is the recommended dosage, Lf is apparently more effective in preterm than in term infants, and the efficacy versus Gram-negative bacteria could be limited. Important gaps in the knowledge exist concerning dosages, schedule, duration of treatment, most effective probiotic strains, and interactions of probiotics with human and bovine milk [12,112]. As stated below, experimental findings in preterm piglets support these claims [113,114].

### 5.2. Piglet Model

Given their physiological and anatomical resemblance to humans, piglets have been used in experimental studies as a model for the neonatal gastrointestinal tract to provide insights about the presumable mechanisms underlying the role of Lf and its derivatives on intestinal homeostasis in neonates [100].

A recombinant Lf fusion peptide consisting of Lfcin and lactoferrampin (Lfampin) expressed in *Pichia pastoris* has been shown to exhibit potent probiotic effects on bifidobacteria and lactobacilli throughout the gastrointestinal tract in weaned piglets [115]. Additional trials in healthy full-term piglets demonstrated that bLf, in combination with probiotics increased the richness of microbiota in the small and large intestine. Interestingly, bLf reversed the effects of probiotics on the increase or decrease of ferric or ferrous iron transport system abundance, respectively [116]. Thus, the iron-binding ability of bLf on ferric ions seems to affect the role of probiotics on microbiota modulation.

Piglet trials documented the substantive effects of bLf or recombinant hLf on stimulating the abundance of a wide array of microbiota members and on improving body weight gain [5,13]. Recombinant Lfcin-Lfampin expressed in *Photorhabdus luminescens* also enhanced the growth of bifidobacteria and lactobacilli and body mass gain [117]. Glycan degradation results in the formation of SFCA, which in turn are a source of energy for the epithelial cells necessary for ongoing gut maturation [110]. Thus, underlying mechanisms seem to be associated with the ability of Lf or its derivatives to favor the diversity of microbiota members with a pivotal role in breakdown glycans.

The microbiome has a critical role in intestinal maturation by providing signals for the development of innervations of the enteric nervous system connected in turn with the central nervous system (CNS); conversely, CNS and enteric nerves modulate intestinal maturation via microbiome signaling pathways [118]. Notably, the effect of bLf on the abundance and diversity of microbiota in piglets seems to affect the intestinal expression of neurotransmitters such as ileal vasoactive intestinal peptide released by enteric nerve fibers [5] and the expression of parameters associated with the maturation of enteric nerves such as brain-derived neurotrophic factor (BDNF) and ubiquitin carboxy-terminal hydrolase 1 [ubiquitin thiol esterase (UCHL1)] [5,14]. These results suggest a presumable mechanism for the role of bLf on the interplay between the gut-brain axis and the microbiome, resulting in the maturation of enteric nerve fibers.

The impact of bLf on intestinal maturation has also been demonstrated with the increased activity of brush-border enzymes, such as jejunal lactase, as found in piglets fed formula containing probiotics and bioactive milk components, including bLf [5]. The intestinal alkaline phosphatase activity was enhanced in piglets fed a sow milk replacement containing bLf [14]. Benefits on epithelial architecture have been evidenced in the small intestine by larger crypt area, depth and width and thinner lamina propria, as found in piglets fed bLf, recombinant hLf or Lfcin-Lfampim from *P. luminescens* [14,117,119]. bLf also enhanced jejune crypt proliferation, depth, area, and the crypt mRNA expression of β-catenin mRNA, as documented in colostrum-deprived piglets fed formula containing bLf [120]. The upregulation of bLf on intestinal growth may result from the elicitation of the β-catenin-Wnt signaling pathway [120]. β-Catenin mRNA encodes a cytosolic protein expressed at crypts that is regarded as a key effector of Wnt signaling; the latter drives the self-renewal and proliferation of stem cells and their concomitant differentiation to other cell components of the epithelial monolayer [121]. The findings provide evidence that supports the role of Lf in growth and maturation in the neonatal intestine.

Maternal bLf supplementation increased the pregnancy rate, litter size, and survival, and the levels of IgA antibodies in the serum of gilts and their litter [122]. These findings indicate that bLf consumption provided benefits on survival and immunity during pregnancy and lactation. In piglets fed daily for the first seven or 14 days of life with a formula containing bLf (367 or 1300 mg/kg body weight), the serum IgG antibodies increased; however, bLf did not affect the cellularity of the lymphoid populations (B cells and T cells), NKCs and neutrophils. In supernatants from lipopolysaccharide (LPS)-primed mesenteric lymph node cell cultures treated with bLf at 367 mg/kg of body mass, the IL-6 and IL-10 levels were enhanced, whereas the tumor necrosis factor (TNF)-α level was unaffected [15]. The consumption of transgenic milk containing hLf had significant effects on the decrease in circulating neutrophils and the increase in lymphocytes without affecting cytokine expression [119], whereas in piglets fed transgenic milk containing recombinant human lysozyme and rhLf, the number of peripheral blood cells was increased without affecting the expression of TNF-α, IL-6, TGF-β and TLR4 [119]. Accordingly, these results indicate that natural or recombinant Lf products display a tendency to either not affect or downmodulate the markers of inflammation.

Studies in models of neonatal piglets have also provided evidence of the modulatory role of bLf and its derivatives on improving the outcome and blunting the inflammation of life-threatening neonatal infectious diseases, including diarrhea and systemic infections [5,14,117,120]. Thus, feeding with bLf, in combination with probiotics, decreased the abundance of opportunistic pathogens [5], while feeding with recombinant Lfcin-Lfampin displayed a selective effect by enhancing the abundance of bifidobacteria and lactobacilli and reducing the intestinal enterotoxigenic *E. coli* [117]. An assessment of spontaneous infections indicated that rhLf-cow milk reduced the incidence of diarrhea [123]. In piglets, bLf had no effect on bifidobacteria and lactobacilli growth, nor did it decrease the frequency and duration of spontaneous early weaning diarrhea [14]. Under the conditions of infection, the results may suggest an interplay between Lf and microbiota on the regulation of the components of innate and adaptive immune responses [117,123]. In piglets infected with *S*. *aureus*, bLf provided a better protective effect than *B*. *infantis* by reducing the severity of the systemic infection. The protective effect of bLf was attributed to the upmodulation of interferon (IFN)-γ and lymphocyte responses resulting in a decrease in bacterial load [120]. Findings indicated that unlike *B*. *infantis*, bLf by itself provided benefits to control neonatal systemic infection by *S*. *aureus*.

By contrast, piglets fed the chimera Lfcin-Lfampin showed enhanced serum antioxidant enzymes such as glutathione peroxidase and peroxidase, as well as the effectors of the adaptive immunity branch including IgA, IgG, and IgM antibodies and the components of the innate response involved in protection from the deleterious effects of inflammation [117]. This finding agrees with the observation that rhLf-cow milk increased TLR-2 mRNA expression in the ileum and the levels of colonic IgG and nuclear factor-κB (NF-κB) p65, concomitant with the increase in spleen IL-2, -4, and -5 and plasmatic IgG, IgA and IL-12, and IL-10 [123]. Although the assessment of parameters was systemic, the compartmental modulatory action on inflammatory and immune components underlies the protective role of recombinant Lf derivatives in the intestinal milieu. According to the above results, some presumable mechanisms that account for the impact of Lf on intestinal microbiota growth are depicted in Figure 3.

Finally, although the potential risks of food allergy have not been found with Lf, as in the case of rhLf [123], the dose of Lf is a critical issue, as described during the development of necrotizing enterocolitis in piglets [113]. In piglets fed formula containing a high dose of bLf (10 g/L), severe enterocolitis and increased colonic permeability were found, whereas IL-1β levels in the proximal small intestine were decreased. To address a presumable mechanism, in vitro assays in porcine intestinal epithelial cells primed with bLf showed that a low bLf dose increased intestinal proliferation and decreased IL-8 generation as well as NF-κB and hypoxia-inducible factor 1α activation, whereas a high bLf dose reversed all of these effects [113]. In vitro assays showed that a low bLf dose enhanced the expression of genes related to energy metabolism and protein synthesis, while a high bLf dose decreased the expression of anti-apoptotic proteins and increased pro-apoptotic proteins and inflammation [114]. According to the above findings, Lf and its derivatives provide beneficial effects on the neonatal intestine by modulating microbiota and by displaying intrinsic regulatory actions on the immune and inflammatory responses. Although Lf shows no toxic effects, special attention should be paid to avoiding the potential risks of high Lf doses.

## 6. Practical Implications

At present, many bacterial species in the human commensal microbiota, apart from lactobacilli and bifidobacteria, may have the potential activity to strengthen the epithelial barrier and alleviate inflammation, opening the possibility of their subsequent use in therapies for the treatment of intestinal inflammatory diseases [124]. These “next-generation probiotics” include bacterial species such as *Akkermansia muciniphila*, *Roseburia intestinalis, Fecalibacterium prausnitzii,* and *Eubacterium hallii*, which are among the most abundant bacteria in the feces of healthy people. These bacteria enhance the ability of some probiotics to contribute to intestinal health through processes that are independent of immunomodulation and are more closely related to the metabolic processes of dietary components [124].

The dual activity of Lf and its derivatives as antimicrobials and modulators of the microbiota and immunity is the focus of intense analysis to be applied as antimicrobial replacements. Pharmaceutical formulations of enteric-coated Lf tablets containing *L. brevis* may be a novel trend to prevent and control human intestinal dysfunctions [125]. Lfcin-Lfampin could be used as a replacement for antibiotic therapy for sepsis, taking advantage of its selective antimicrobial action and probiotic effect as an immune modulator [115,117]. Recombinant *Lactobacillus* strains expressing Lf [126,127] may be future strategies to prevent and combat high-risk human infections, such as neonatal necrotizing enterocolitis [128]. As documented in murine models and clinical trials, pharmaceutical preparations containing bLf and probiotics or Lf-producing probiotic strains have been effective to combat vulvovaginal infections caused *Candida albicans*, *Gardnerella vaginalis,* or *Atopobium vaginalis* by restoring the microbiota in vulvovaginal tract [129,130,131]. These findings provide promising insights to control vulvovaginal infections caused resistant pathogens via microbiota modulation.

The potential application of bovine lactoferrin in combination with probiotic bacteria in pharmaceutical products formulated to enhance and strength the host mechanisms for the restoration of intestinal homeostasis could be a sustainable strategy to reduce the use of antibiotics against multidrug-resistant microorganisms causing life-threatening infections in neonates. Future investigations are needed to provide insights about the substantive effects of Lf and derivatives on the intestinal wellness through its modulatory effect on intestinal inhabitant bacteria and the limitations and potential drawback of their use.

## 7. Conclusions

The use of antimicrobials in human and veterinary medicine has led to an exponential increase in selected multidrug-resistant organisms that are highly refractory to antibiotic treatment and to protracted infectious diseases that threaten health and life. This very serious problem has prompted the search for strategies to prevent and control infectious diseases using therapeutic methods based on natural compounds and synthetic derivatives produced with sustainable technologies. From the data reported in this review, it is evident that the immunomodulation of the intestinal immune system by Lf on probiotics may provide natural and sustainable approaches to control infectious diseases by strengthening intestinal homeostasis rather than by combating pathogenic microorganisms with antibiotics.

## Figures and Tables

**Figure 1 ijms-20-04707-f001:**
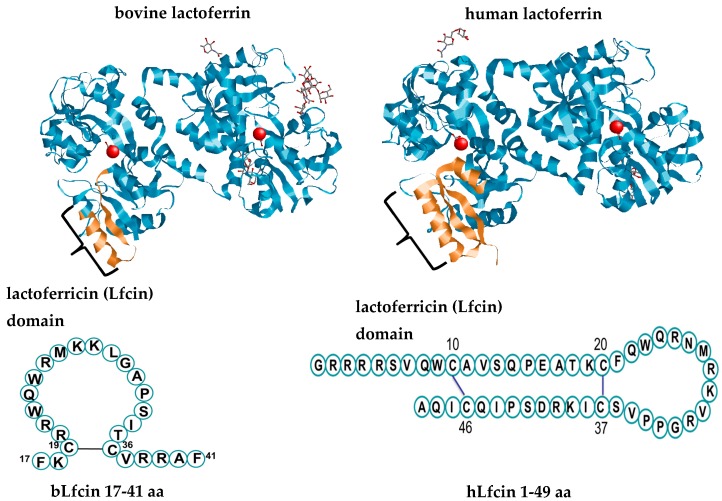
The canonical structure of lactoferrin (Lf) (bovine lactoferrin Protein Data Bank (PDB) Identification (ID):1BLF; human lactoferrin PDB ID: LFG), consisting of a single polypeptide chain organized in N- and C-lobes that in turn contain N1/N2 and C1/C2 subdomains. Each subdomain forms a deep cleft in which one ferric ion (Fe^3+^) and one bicarbonate ion (HCO_3_^−1^) are bound. The N1-terminal subdomain contains the lactoferricin (Lfcin) domain, which is released in free form following the proteolysis of the full-length Lf with pepsin. The primary sequences of bovine (PDB ID: 1LFC) and human (PDB ID: 1Z6V) Lfcin are depicted below the full-length Lf. The molecular modeling and coloring of the tertiary structures were accomplished with the free-download software RasMol™ 1994, designed and launched by Roger Sayle©.

**Figure 2 ijms-20-04707-f002:**
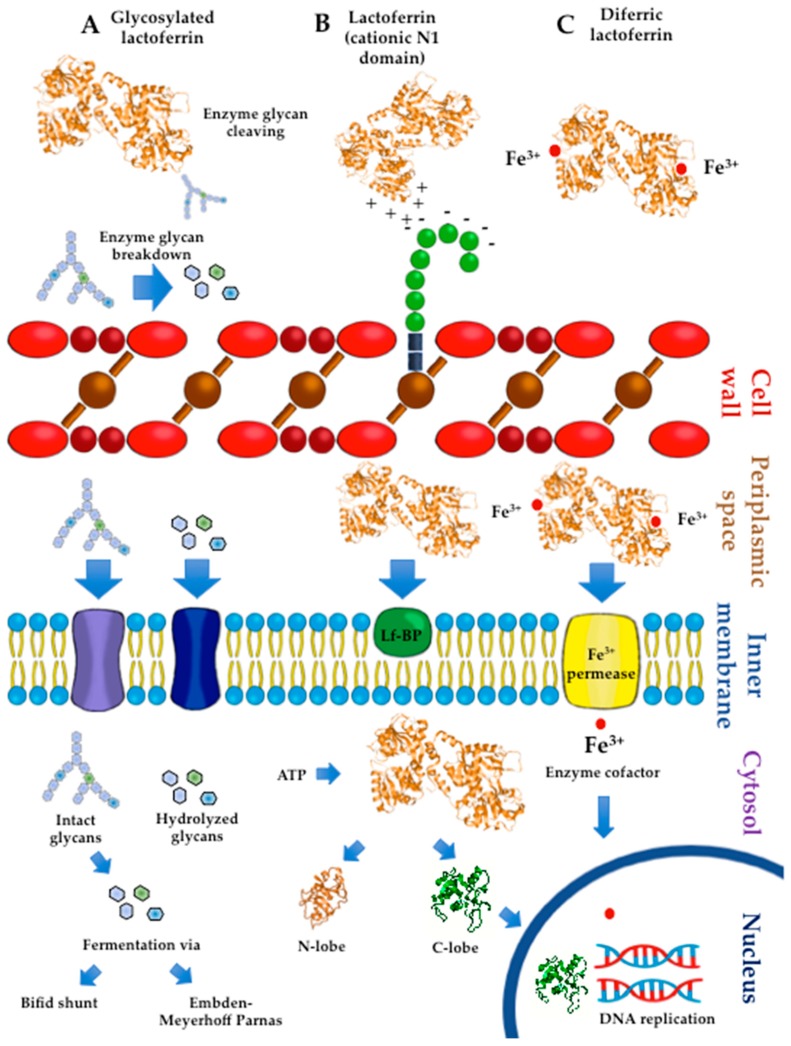
Simplified hypothetical mechanisms that may account for the in vitro growth-promoting effects of lactoferrin (Lf) on probiotics (bifidobacteria and lactobacilli). (**a**) Lf glycans undergo enzymatic hydrolysis to produce both intact oligosaccharides and the products of cleavage (disaccharides and monosaccharides). Intact and hydrolyzed glycans are transported via cell membrane permeases to the cytosol. In the cytosol, glycans are terminally hydrolyzed to yield monosaccharides used by cytosolic enzymes as substrates of fermentation via the Embden–Meyerhof–Parnas or bifid shunt pathways to produce energy for cell growth. (**b**) The cationic surface of Lf interacts electrostatically with acidic components of the cell wall that are negatively charged, such as teichoic acids. This interaction may favor the internalization of Lf in the periplasmic space and its eventual binding to cell membrane proteins for translocation to the cytosol. In the presence of ATP, Lf dissociates to the N-lobe and C-lobe. The latter is internalized in the nucleus and interacts with DNA to modulate genes involved in the mechanisms of DNA replication for cell growth. (**c**) Diferric Lf reaches the periplasmic space, where iron in ferric form is released and then translocated via a permease to cytosolic compartments. Ferric ions are considered an essential cofactor of enzymes involved in crucial processes of metabolism and DNA replication for growth. RasMol™ 1994 by Roger Sayle© was used as the software for modeling and coloring the tertiary protein structures.

**Figure 3 ijms-20-04707-f003:**
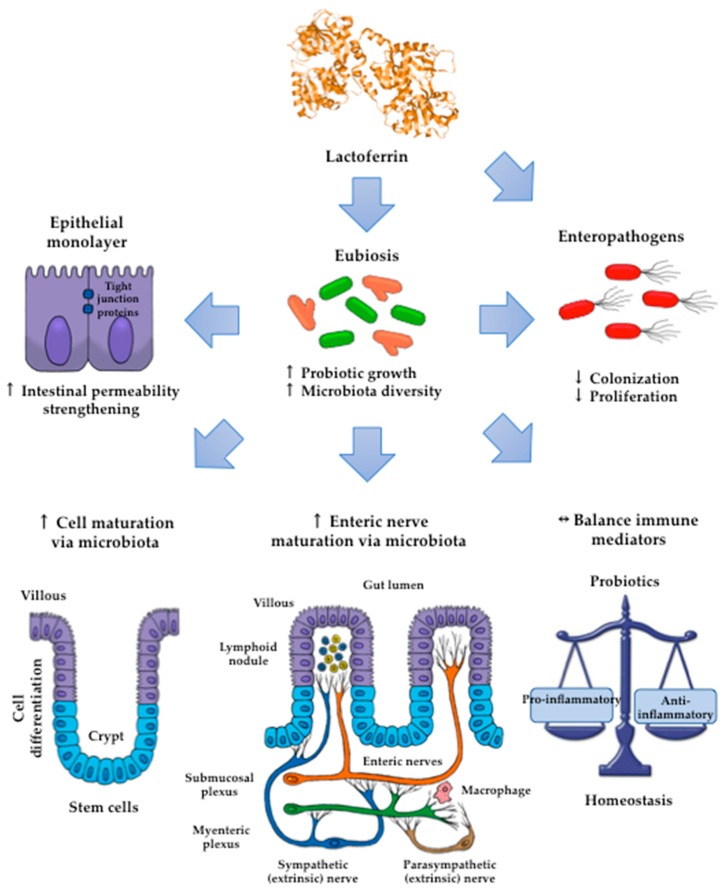
The impact of Lf on the promotion of the growth and diversity of intestinal microbiota may entail (i) strengthening of the permeability of the epithelial cell monolayer; (ii) favoring of the microbial antagonism that discourages the colonization and proliferation of enteric pathogens, enhancing the growth and maturation of (iii) cell-monolayer components and (iv) gut nerve fibers; and (v) providing signals to balance the anti- and proinflammatory responses resulting in homeostasis.

**Table 2 ijms-20-04707-t002:** Fluctuations on lactoferrin levels in postpartum mammary gland secretions.

Human [28]		Bovine [29]	
Days postpartum	hLf mg/mL	Weeks postpartum	bLf mg/mL
0–5 colostrum	5.05	0−1 colostrum	0.7328
5-15	3.30	2	0.6047
16-30	2.31	4	0.5411
31-60	1.95	6	0.4027
61-90	1.89	8	0.3503

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
