# Peer review of "The Impact of Lactoferrin on the Growth of Intestinal Inhabitant Bacteria"

_ijms, 2019, doi:10.3390/ijms20194707_

Round 1

Reviewer 1 Report

Very interesting work and well organized.

Lines 63-65. Delete

Line 79. Table 1 must be placed here, after its first mention in the text. Same for fig 1.

Lines 116-316.  Section 3. Probiotics and the modulation of the gut immune response should be deleted. It is not the topic of the present review. It is very general knowledge. It is characteristic that not even once the Lf is reported here.

Critical comments by the authors are missing. Probably more critical discussion is needed in the section 6. Practical implications.

Several recent studies are missing. Please add recent references in your review and discuss them:

1. Nazir, S., Nasir, M., Yasmeen, A., & Usman, S. (2017). Review study on lactoferrin: A multifunctional protein. Sky Journal of Food Science, 6(2), 014-020.

2. Liao, H., Liu, S., Wang, H., Su, H., & Liu, Z. (2019). Enhanced antifungal activity of bovine lactoferrin-producing probiotic Lactobacillus casei in the murine model of vulvovaginal candidiasis. BMC microbiology, 19(1), 7.

3. Russo, R., Superti, F., Karadja, E., & De Seta, F. (2019). Randomised clinical trial in women with Recurrent Vulvovaginal Candidiasis: Efficacy of probiotics and lactoferrin as maintenance treatment. Mycoses, 62(4), 328-335.

4. Manzoni, P. (2016). Clinical benefits of lactoferrin for infants and children. The Journal of pediatrics, 173, S43-S52.

5. Russo, R., Karadja, E., & De Seta, F. (2019). Evidence-based mixture containing Lactobacillus strains and lactoferrin to prevent recurrent bacterial vaginosis: a double blind, placebo controlled, randomised clinical trial. Beneficial microbes, 10(1), 19-26.

6. Kim, W. S. (2019). Effects of Heat-treated Bovine Lactoferrin on the Growth of Lactococcus lactis subsp. cremoris JCM 20076. Journal of Milk Science and Biotechnology, 37(2), 129-135.

Author Response

Comments and Suggestions for Authors

Very interesting work and well organized.

Thank so much for your comment

Lines 63-65. Delete

Done

Line 79. Table 1 must be placed here, after its first mention in the text. Same for fig 1.

Done

Lines 116-316.  Section 3. Probiotics and the modulation of the gut immune response should be deleted. It is not the topic of the present review. It is very general knowledge. It is characteristic that not even once the Lf is reported here.

Critical comments by the authors are missing. Probably more critical discussion is needed in the section 6. Practical implications.

Section 3 was not deleted instead of it was shortened taking into account the suggestion of the reviewer 3. In the current version section 3 contains essential background of aspects related with the content of the section 4 and 5.

Critical comments were included in necessary case and also in the section 6 corresponding to Practical implications.

Several recent studies are missing. Please add recent references in your review and discuss them:

As indicated by the referee, all the references were added and discussed in the current version.

Nazir, S., Nasir, M., Yasmeen, A., & Usman, S. (2017). Review study on lactoferrin: A multifunctional protein. Sky Journal of Food Science, 6(2), 014-020. Liao, H., Liu, S., Wang, H., Su, H., & Liu, Z. (2019). Enhanced antifungal activity of bovine lactoferrin-producing probiotic Lactobacillus casei in the murine model of vulvovaginal candidiasis. BMC microbiology, 19(1), 7. Russo, R., Superti, F., Karadja, E., & De Seta, F. (2019). Randomised clinical trial in women with Recurrent Vulvovaginal Candidiasis: Efficacy of probiotics and lactoferrin as maintenance treatment. Mycoses, 62(4), 328-335. Manzoni, P. (2016). Clinical benefits of lactoferrin for infants and children. The Journal of pediatrics, 173, S43-S52. Russo, R., Karadja, E., & De Seta, F. (2019). Evidence-based mixture containing Lactobacillus strains and lactoferrin to prevent recurrent bacterial vaginosis: a double blind, placebo controlled, randomised clinical trial. Beneficial microbes, 10(1), 19-26. Kim, W. S. (2019). Effects of Heat-treated Bovine Lactoferrin on the Growth of Lactococcus lactis subsp. cremoris JCM 20076. Journal of Milk Science and Biotechnology, 37(2), 129-135.

Submission Date

10 August 2019

Date of this review

13 Aug 2019 07:56:54

Reviewer 2 Report

The manuscript by Vega-Bautista et al. reviews the basic biochemistry of lactoferrin and its functions and impact on the gut microbiota and immune responses. This review paper provides useful information and perspective in the field of lactoferrin research and is worthy of dissemination.

Minor Comments:

1. The authors should define the timeliness of the reviewed references in Introduction section, which would help readers understand the latest progress of the field.

2. Can the authors comment more on mechanisms of Lf against MRSA or multidrug-resistant microorganisms?

3. Some abbreviations (e.g., apo-/holo-Lf) should be clearly defined.

4. Table 2, it’s better to keep consistent of the unit (mg/mL or ug/mL) for comparison purposes.

Author Response

Comments and Suggestions for Authors

The manuscript by Vega-Bautista et al. reviews the basic biochemistry of lactoferrin and its functions and impact on the gut microbiota and immune responses. This review paper provides useful information and perspective in the field of lactoferrin research and is worthy of dissemination.

Minor Comments:

The authors should define the timeliness of the reviewed references in Introduction section, which would help readers understand the latest progress of the field.

As indicated by the reviewer, the timeliness was stated explicitly in the introduction section.

Can the authors comment more on mechanisms of Lf against MRSA or multidrug-resistant microorganisms?

As indicated by the reviewer, mechanisms of Lf against MRSA or multidrug resistant microorganisms were stated in the current version (see please section 4.2 and 4.5)

Some abbreviations (e.g., apo-/holo-Lf) should be clearly defined.

Apo-/holo Lf were defined in lines 78-79 section 2 of the first version. In the current version all acronyms were clearly defined before applying them in the text body.

Table 2, it’s better to keep consistent of the unit (mg/mL or ug/mL) for comparison purposes.

As indicated by the referee, in the Table 2, data were expressed in the same units.

Submission Date

10 August 2019

Date of this review

20 Aug 2019 20:43:50

Reviewer 3 Report

Ijms-581608: The impact of lactoferrin on the growth of intestinal inhabitant bacteria

In this review, Vega-Bautista et al. discuss the effect on lactoferrin (Lf), a milk glycoprotein, on the growth of probiotics and influence on the intestinal microbiota. They include both in vitro and in vivo studies, comparing human Lf and bovine Lf, and looking at studies in human and pig models. Overall the document is easy to follow. The figures and tables are clear and contribute to the document as a whole.

Specific Comments

Lines 47-49: Please reword this sentence for clarity (especially the beginning). The way it is worded the authors state that Lf and Lfcins do not affected the growth of gut-beneficial microorganisms and that they inhibit some gut-beneficial microorganisms. Additionally, in lines 52-53 the authors talk about Lf impacting growth of intestinal microbiota, making it confusing to summarize what has clearly been found in studies looking at Lf and the growth of the intestinal microbiota.

Lines 63-65: Please remove the description of what the section should contain.

Line 66: Are the authors citing Lf’s DNA or protein sequence similarity percentage?

Section 3: (Lines 116-316): This section, while clearly written, is long and detailed, and most of the details discussed are not talked about/linked to subsequent sections dealing with Lf. The authors should shorten this section and remove detail to only give background that is needed for understanding of Lf and the growth of probiotic bacteria (Section 4) and Lf’s modulatory effects on the gut microbiota (Section 5). For example, section 3.1 (General Properties) is about the breakdown of oligosaccharides in milk in great detail, whereas general properties would list a wide variety of what probiotics do.

Line 368: Please provide the appropriate reference(s) for the MRSA finding.

Author Response

Comments and Suggestions for Authors

Ijms-581608: The impact of lactoferrin on the growth of intestinal inhabitant bacteria

In this review, Vega-Bautista et al. discuss the effect on lactoferrin (Lf), a milk glycoprotein, on the growth of probiotics and influence on the intestinal microbiota. They include both in vitro and in vivo studies, comparing human Lf and bovine Lf, and looking at studies in human and pig models. Overall the document is easy to follow. The figures and tables are clear and contribute to the document as a whole.

Thank you for your comments

Specific Comments

Lines 47-49: Please reword this sentence for clarity (especially the beginning). The way it is worded the authors state that Lf and Lfcins do not affected the growth of gut-beneficial microorganisms and that they inhibit some gut-beneficial microorganisms.

Thank you for this critical remark. The alluded paragraph was re-phased in the current version.

Additionally, in lines 52-53 the authors talk about Lf impacting growth of intestinal microbiota, making it confusing to summarize what has clearly been found in studies looking at Lf and the growth of the intestinal microbiota.

Thank you for this critical remark. The alluded paragraph was re-phased in the current version.

Lines 63-65: Please remove the description of what the section should contain.

Statement (lines 63-65) was removed as requested by the referee

Line 66: Are the authors citing Lf’s DNA or protein sequence similarity percentage?

Thank you for this critical remark. Statement refers to protein sequence as defined in the current text.

Section 3: (Lines 116-316): This section, while clearly written, is long and detailed, and most of the details discussed are not talked about/linked to subsequent sections dealing with Lf. The authors should shorten this section and remove detail to only give background that is needed for understanding of Lf and the growth of probiotic bacteria (Section 4) and Lf’s modulatory effects on the gut microbiota (Section 5). For example, section 3.1 (General Properties) is about the breakdown of oligosaccharides in milk in great detail, whereas general properties would list a wide variety of what probiotics do.

As indicated by the referee this section was shortened in the current version and contains background information of aspects related with the content of section 4 and 5.

Line 368: Please provide the appropriate reference(s) for the MRSA finding.

Done

Submission Date

10 August 2019

Date of this review

21 Aug 2019 20:25:04

Round 2

Reviewer 1 Report

The work has been improved according to reviewers comments.
